# A Quasi-Velocity-Based Tracking Controller for a Class of Underactuated Marine Vehicles

## Przemyslaw Herman

Institute of Automatic Control and Robotics, Poznan University of Technology, ul. Piotrowo 3a, 60-965 Poznan, Poland; przemyslaw.herman@put.poznan.pl; Tel.: +48-61-224-4500

**Abstract:** This paper investigates the trajectory tracking control problem for underactuated underwater vehicles, for which a model is expressed in terms of quasi-velocities arising from the inertia matrix decomposition. The control approach takes into account non-modeled dynamics and external disturbances and is suitable for symmetric vehicles. It is shown that such systems can be diagonalized using inertial quasi-velocities (IQVs). The strategy consists of the velocity controller and two adaptive integral sliding mode control algorithms. The proposed approach, introducing velocity transformation and using backstepping methods and integral sliding mode control, allows trajectory tracking for vehicles in described models with symmetric inertia matrix. Proof of the stability of the closed system was carried out using IQV. The proposed scheme has been verified on two 3 DOF models of underwater vehicles with thruster limitations. A brief discussion of the results is also given.

**Keywords:** underactuated marine vehicle; trajectory tracking; backstepping; integral sliding mode control; robustness; velocity transformation; quasi-velocities; simulation





## 1. Introduction

The control of marine vehicles is an area where active research is being conducted due to the various theoretical challenges and significant practical applications of these vehicles. Most marine vehicles are underactuated, which means they have more degrees of freedom to control than the number of independent control inputs.

Particularly interesting and challenging among the control problems studied for marine vehicles are trajectory tracking and path tracking. The latter issue is particularly relevant for mobile vehicles and robots, e.g., [1–3]. In [1], the predictive Stanley controller was applied to path tracking. The authors of [2] has presented a robust path tracking algorithm under the vehicle-infrastructure cooperative system (i-VICS) based on Kalman filter. Another article [3] proposed a navigation method modeled on the use of bookmarks as in the memory of living beings to reach a destination using key points.

In general, an underwater vehicle is modeled with 6 degrees of freedom (DOF) and control schemes are designed for such a model, as can be found in [4–6]. However, in some works, the dynamics model is reduced to 5 DOF and only the controller is designed for it. Such an approach is presented, e.g., in [7,8].

However, this paper only addresses the problem of tracking the desired trajectory for a 3 DOF vehicle model in horizontal motion. In the simplest case, the inertia matrix is assumed to be diagonal. There are various control schemes that have been successfully applied to this class of vehicles. One can point to works that use sliding mode control (SMC), such as [9,10]. Another group of controllers is based on Lyapunov method [11,12]. Control strategies based on the backstepping approach are also widely used [13–15]. This technique is often combined with other methods, such as SMC [16,17], direct Lyapunov method [18,19], averaging approach [20], dynamic surface control method [21], nonlinear model predictive control (NMPC) [22], neural network (NN) and SMC [23,24]. Another group of methods uses NN, such as in [25]. Such approaches are also a combination

of different algorithms, such as with SMC [26] or with fuzzy logic [27]. Fuzzy logic-based control schemes are also available for solving trajectory tracking problems, for example [28,29]. Many control algorithms based on other methods can be found, such as event-triggered control (ETC) with fuzzy system approximation [30] or with NN [31] as well as proportional integral derivative sliding mode control (PID-SMC) [32] or with prescribed performance [33].

Models with a diagonal inertia matrix allow simplification of the controller, but from the point of view of dynamics they often deviate from the real vehicle. For this reason, some control schemes are designed based on a model with a symmetric inertia matrix. Many approaches to the issue under consideration can be identified here. The NN has been used, for example, in [34,35], although additional methods such as backstepping in [36] or backstepping and SMC in [37] are sometimes applied. Various approaches can be cited here, such as those based on the backstepping method [38], SMC [39] terminal sliding mode control [40], prescribed performance [41] or input–output linearization [42]. For this type of model, control strategies that are a combination of different methods are also applied, such as backstepping or the Lypunov approach [43].

Some control algorithms are based on the transformation of variables. In such cases, the control scheme is designed in new variables. One possible solution is to introduce coordinate transformation as in [14,29] for a model with a diagonal inertia matrix. The coordinate transformation is also used for models with a non-diagonal inertia matrix as in [41,42,44].

The proposed control scheme for tracking the trajectory of 3 DOF underactuated underwater vehicles that move in the horizontal plane is based on a velocity transformation instead of a coordinate transformation. The controller uses the inertial quasi-velocity (IQV) originally described for mechanical systems in [45]. Because of the difficulties associated with the velocity transformation (since there is additional dynamic coupling due to the coupling between these velocities), controllers incorporating the IQV are used in systems with fully actuated input signals, as can be found in [46,47] (for serial manipulators) or [48,49] (for fully actuated marine vehicles). This means that for vehicles with incomplete input signals, developing control algorithms in terms of IQV is still a difficult problem. The present work is an attempt to fill this research shortfall.

An important advantage of controllers based on IQV is the availability of information about the impact of vehicle dynamics during the control task. This means that, in addition to trajectory tracking, it is possible to gain insight into the dynamics of the system, and thus study the behavior of the vehicle with the controller depending on the assumed desired trajectory, as well as changes in the model parameters. An example of such studies for a system with full forcing is given in [49]. Thus, the novelty of the work is also due to the fact that IQV cannot be easily applied to mechanical systems with underactuation. Simulation results show effectiveness of the proposed control scheme and give an answer to the question of the usability of using QV for trajectory tracking control of underactuated underwater vehicles.

Paper [50] dealt with an algorithm based on a velocity transformation but also a coordinate transformation, while this one uses only the first transformation. The dynamic models are different in the two works and the disturbances are defined differently. The algorithm expressed in IQV [50] was only a slight modification of the original controller while here the control scheme is based on a combination of IQV, backstepping and SMC, i.e., it is grounded on a different control idea.

The main contributions of this study are:

(i)　Development of a trajectory tracking controller in terms of IQV for asymmetric underactuated vehicles moving in the horizontal plane.

(ii)　Extension of IQV concept for tracking controller for underactuated vehicles with 3 DOF.

(iii) Compared to existing controllers employing a combination of bakstepping and SMC methods, the proposed algorithm uses a velocity transformation resulting in a vector of new variables that includes the dynamic parameters of the vehicle model.

(iv) Indication of the mathematical conditions for the implementation of such an algorithm and its verification based on simulation studies performed for two vehicles with different dynamics and for two different desired trajectories.

*Comment.* Verification of the control algorithm for models with different parameters, as well as the use of two trajectories and an intuitive method of selecting controller gains seems reasonable, as it can happen that such changes affect the performance of the controller, as shown, for example, in [51].

The present work is organized as follows. Section 2 presents the formulation of the problem. Section 3 describes the equations of motion in terms of quasi-velocities. Section 4 provides a theoretical analysis of the proposed control scheme. Section 5 presents numerical results for two different underwater vehicles and two trajectories to demonstrate the effectiveness of the controller. Section 6 discusses the obtained results in comparison with other works on the issue under consideration. Finally, Section 7 presents conclusions.

## 2. Problem Formulation

Consider the model of marine vehicle moving horizontally given in Figure 1.

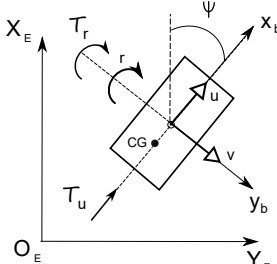

**Figure 1.** Marine vehicle model sketch.

For the position and orientation description, the North–East–Down (NED) frame is applied, namely $\eta = [x, y, \psi]^T$. The velocities in the body frame are defined by $v = [u, v, r]^T$ which means that the surge velocity, the sway velocity, and the yaw velocities are taken into account. The equations of motion for a marine vehicle moving horizontally are as follows [4,5]:

$$\dot{\eta} = R(\psi)v, \tag{1}$$

$$M\dot{v} + C(v)v + D(v)v = \tau + f_E, \tag{2}$$

where $R(\psi)$ is the rotation matrix, $M$ is the symmetric inertia matrix which contains additional mass (the matrix is positive definite), $C(v)$ is the matrix of Coriolis and centripetal terms, and $D(v)$ is the damping matrix. Moreover, $\tau = [\tau_u, 0, \tau_r]^T$, where $\tau_u$ is the thruster force and $\tau_r$ is the yaw torque, $f_E$ means the vector of external disturbances. The corresponding matrices and vectors are of the form:

$$R(\psi) = \begin{bmatrix} \cos\psi & -\sin\psi & 0 \\ \sin\psi & \cos\psi & 0 \\ 0 & 0 & 1 \end{bmatrix}, \quad M = \begin{bmatrix} m_{11} & 0 & 0 \\ 0 & m_{22} & m_{23} \\ 0 & m_{23} & m_{33} \end{bmatrix},$$

$$C(v) = \begin{bmatrix} 0 & 0 & c_{13} \\ 0 & 0 & c_{23} \\ -c_{13} & -c_{23} & 0 \end{bmatrix}, \quad D(v) = \begin{bmatrix} d_{11} & 0 & 0 \\ 0 & d_{22} & d_{23} \\ 0 & d_{32} & d_{33} \end{bmatrix}, \quad f_E = \begin{bmatrix} f_{Eu} \\ f_{Ev} \\ f_{Er} \end{bmatrix} \tag{3}$$

with: $m_{11} = m - X_{\dot{u}}$, $m_{22} = m - Y_{\dot{v}}$, $m_{23} = m_{32} = mx_g - Y_{\dot{r}}$, $m_{33} = J_z - N_{\dot{r}}$, $c_{13} = -m_{22}v - m_{23}r$, $c_{23} = m_{11}u$, $d_{11} = X_u + X_{|u|u}|u|$, $d_{22} = Y_v + Y_{|v|v}|v| + Y_{|r|v}|r|$, $d_{23} = Y_r + Y_{|v|r}|v| + Y_{|r|r}|r|$, $d_{32} = N_v + N_{|v|v}|v| + N_{|r|v}|r|$, and $d_{33} = N_r + N_{|v|r}|v| + N_{|r|r}|r|$.

## 3. Equations of Motion in Terms of Quasi-Velocities

To use the decomposition method, one must assume that the matrix $M$ is symmetric. It is possible to meet this assumption as long as one takes into account all model inaccuracies and external disturbances in the disturbance function with limitations on its values, as will be shown in the next section. The $M$ symmetric matrix can be decomposed using one of the known methods, such as [45] which was successfully applied for marine vehicles, for example in [49] (Equation (1) is valid). This method was developed for mechanical systems [45]. It is easy to implement [49] and has low computational complexity which reduces simulation time. In addition, it allows physical interpretation of IQV, and the physical units are the same as in classical equations of motion. The inertia matrix is factorized as $M = Y^{-T}N_{diag}Y^{-1}$ which leads to a diagonal matrix $N_{diag} = Y^T M Y$. However, since the control algorithm must use known quantities, the decomposed matrix is of the form $N = \hat{Y}^T M \hat{Y}$ (the matrix $\hat{Y}$ contains nominal parameters), so the resulting errors $\Delta Y$ are included in the new disturbance vector which is defined as $f = f_E + \Delta Y$. Then, the kinematic Equation (1) remains unchanged, but instead of (2) one obtains:

$$N\dot{\zeta} + \hat{Y}^T C(\nu)\nu + \hat{Y}^T D(\nu)\nu = \hat{Y}^T \tau + \hat{Y}^T f, \tag{4}$$

$$\nu = \hat{Y}\zeta, \tag{5}$$

$$\hat{Y} = \begin{bmatrix} 1 & 0 & 0 \\ 0 & 1 & \hat{Y}_{23} \\ 0 & 0 & 1 \end{bmatrix}, \quad N = \text{diag}\{N_1, N_2, N_3\}, \tag{6}$$

where the new vector of quasi-velocities $\zeta = [\zeta_1, \zeta_2, \zeta_3]^T$. The introduced quantities are defined as: $N_1 = m_{11}$, $N_2 = m_{22}$, $N_3 = m_{33} - (m_{23}^2/m_{22})$, $\hat{Y}_{23} = -(\hat{m}_{23}/\hat{m}_{22})$.

Equations of motion replacing (4) and (5) may be written as:

$$\zeta_1 = u, \tag{7}$$

$$\zeta_2 = v - \hat{Y}_{23}r, \tag{8}$$

$$\zeta_3 = r, \tag{9}$$

$$N_1\dot{\zeta}_1 = F_1(\zeta) + \tau_u + f_u, \tag{10}$$

$$N_2\dot{\zeta}_2 = F_2(\zeta) + f_v, \tag{11}$$

$$N_3\dot{\zeta}_3 = F_3(\zeta) + \tau_r + f_{\zeta_3}, \tag{12}$$

where $f_{\zeta_3} = \hat{Y}_{23}f_v + f_r$. This means that only $v \neq \zeta_2$ because $v = \zeta_2 + \hat{Y}\zeta_3$. In order to avoid a decomposition that would result in unknown quantities, the original variables were used where they are present, namely:

$$F_1(\zeta) = (m_{22}v + m_{23}r)r - d_{11}u, \tag{13}$$

$$F_2(\zeta) = -m_{11}ur - (d_{22}v + d_{23}r), \tag{14}$$

$$F_3(\zeta) = -(m_{22}v + m_{23}r)u + m_{11}uv - \hat{Y}_{23}m_{11}ur - (\hat{Y}_{23}d_{22} + d_{32})v - (\hat{Y}_{23}d_{23} + d_{33})r. \tag{15}$$

## 4. Trajectory Tracking Control Algorithm

The proposed controller in terms of the IQV enables the trajectory tracking in the horizontal plane taking into account parameter perturbations in the presence of external disturbances.

### 4.1. Tracking Problem and Assumptions

The desired trajectory is expressed as $\eta_d = [x_d, y_d, \psi_d]^T$ whereas the tracking errors are $[x_e^E, y_e^E, \psi_e]^T = [x - x_d, y - y_d, \psi - \psi_d]^T$. The functions $x_d, y_d$ defined in the reference frame

$\{E\}$ must be smooth and continuous. The desired attitude angle trajectory is determined from the desired trajectory from the Equation (such as in [15]):

$$\psi_d = \arctan\left(\frac{\dot{y}_d}{\dot{x}_d}\right). \tag{16}$$

Moreover, the coordinate transformation is applied:

$$x_e = \cos\psi\, x_e^E + \sin\psi\, y_e^E \tag{17}$$
$$y_e = -\sin\psi\, x_e^E + \cos\psi\, y_e^E, \tag{18}$$

because $x_e$, $y_e$ are defined in the body frame $\{B\}$. Calculating the time derivative of the kinematic tracking errors and making use of (1), the following can be written:

$$\dot{x}_e = u - u_d\cos\psi_e + ry_e \tag{19}$$
$$\dot{y}_e = v + u_d\sin\psi_e - rx_e, \tag{20}$$
$$\dot{\psi}_e = r - r_d, \tag{21}$$

where $u_d = \sqrt{\dot{x}_d + \dot{y}_d}$ and $r_d = \dot{\psi}_d$ mean the reference velocities.

**Assumption 1.** *The model parameter perturbations are bounded, i.e., $|m_{ij} - \hat{m}_{ij}| \le \tilde{m}_{ij}$, $|d_{ij} - \hat{d}_{ij}| \le \tilde{d}_{ij}$ where $i, j = 1, 2, 3$, $|N_3 - \hat{N}_3| \le \tilde{N}_3$, and $\hat{\cdot}$ means the nominal value of the real parameter. This means that there exists the upper bound of the parameter perturbation. Moreover, all the vehicle states are measurable and can be applied in the controller.*

**Assumption 2.** *The external disturbances $f_u$, $f_v$, $f_r$ are unknown but bounded (as, e.g., in [52,53]). This is also true for their first time derivatives [53]. The disturbances may be constant or time-varying.*

**Assumption 3.** *In order to avoid chattering during the vehicle motion under the proposed controller, the discontinuous function signum is approximated by the hyperbolic tangent function, i.e., $sgn(S_i) \approx tanh(\gamma_i S_i)$ and $\gamma_i$ is a positive scalar where $i = 1, 2$ [9].*

**Remark 1.** *The control strategy consists of two control algorithms. The first is used to stabilize tracking errors, while the second is used to move the vehicle to the desired trajectory.*

### 4.2. Kinematic Control Algorithm

In order to design the kinematic controller the following Lyapunov function candidate is proposed, e.g., [54]:

$$V_k = \frac{1}{2}x_e^2 + \frac{1}{2}y_e^2 + (1 - \cos\psi_e). \tag{22}$$

Its time derivative, after using (19)–(21), has the form:

$$\dot{V}_k = x_e\dot{x}_e + y_e\dot{y}_e + \dot{\psi}_e\sin\psi_e = (u - u_d\cos\psi_e)x_e + (r - r_d + u_dy_e)\sin\psi_e + vy_e. \tag{23}$$

Next, because the velocities $u$, $r$ are understood as the virtual control variables to ensure that $\dot{V}_k < 0$, then the desired velocities are proposed as [8,55]:

$$v_{du} = u_d\cos\psi_e - k_x x_e, \tag{24}$$
$$v_{dr} = r_d - u_d y_e - k_\psi\sin\psi_e, \tag{25}$$

where $k_x > 0$, $k_\psi > 0$ are some gain coefficients. If the virtual variables are equal to the desired variables, then substituting the Equations (24) and (25) into the expression (23), the following can be written:

$$\dot{V}_k = -k_x x_e^2 - k_\psi \sin^2 \psi_e + v y_e \leq -k_x x_e^2 - k_\psi \sin^2 \psi_e + |v y_e|. \tag{26}$$

The following lemma is proposed because the direct control in sway direction cannot be ensured.

**Lemma 1.** *The velocity $v$ is bounded if the disturbances $f_v$ are bounded and consequently also the quasi-velocity $\zeta_2$ is limited.*

**Proof.** The Lyapunov function candidate is as follows:

$$V_{\zeta_2} = \frac{1}{2}\zeta_2^2. \tag{27}$$

Since the drag coefficients are dependent on the velocity $v$ and the equation of motion (11) in IQV, to make the analysis easier it can be transformed and expressed by the formula:

$$\dot{\zeta}_2 = N_2^{-1}(-m_{11}ur - d_{22}v - d_{23}r + f_v). \tag{28}$$

The time derivative of (27) can be given in the form:

$$\dot{V}_{\zeta_2} = \zeta_2 \dot{\zeta}_2 = \zeta_2 N_2^{-1}(-m_{11}ur - d_{22}v - d_{23}r + f_v). \tag{29}$$

Moreover, the drag coefficients can be given in the form $d_{22} = Y_{22}(r) + Y_{|v|v}|v|$, $d_{23} = Y_{23}(r) + Y_{|v|r}|v|$ where $Y_{22}(r) = Y_v + Y_{|r|v}|r|$ and $Y_{23}(r) = Y_r + Y_{|r|r}|r|$. Therefore, using the above and (8) it can be written that:

$$\begin{aligned}
\dot{V}_{\zeta_2} &= \zeta_2 N_2^{-1}(-m_{11}ur - d_{22}v - d_{23}r + f_v) \\
&= N_2^{-1}(v - \hat{Y}_{23}r)\big(-m_{11}ur - (Y_{22}(r) + Y_{|v|v}|v|)v - (Y_{23}(r) + Y_{|v|r}|v|)r + f_v\big). \tag{30}
\end{aligned}$$

Let $A = -m_{11}ur - Y_{23}(r)r - Y_{|v|r}|v|r + f_v$. The function $\dot{V}_{\zeta_2} < 0$ if the following conditions are fulfilled:

(1)   $v < \hat{Y}_{23}r < 0$   and   $(A - Y_{22}(r)v - Y_{|v|v}|v|v) > 0$,   and   $(v - \hat{Y}_{23}r) < 0$,    (31)

(2)   $\hat{Y}_{23}r < v < 0$   and   $(A - Y_{22}(r)v - Y_{|v|v}|v|v) < 0$,   and   $(v - \hat{Y}_{23}r) > 0$,    (32)

(3)   $v > \hat{Y}_{23}r > 0$   and   $(A - Y_{22}(r)v - Y_{|v|v}|v|v) < 0$,   and   $(v - \hat{Y}_{23}r) > 0$,    (33)

(4)   $0 < v < \hat{Y}_{23}r$   and   $(A - Y_{22}(r)v - Y_{|v|v}|v|v) > 0$,   and   $(v - \hat{Y}_{23}r) < 0$.    (34)

Taking into account boundedness of $u$ and $r$ (they are controlled and bounded) and the disturbance function $f_v$ bounded by condition (31)–(34) one obtains:

$$(1), \ (2) \quad -v_{max} \leq v < \frac{Y_{22}(r) - \sqrt{\Delta_1}}{2Y_{|v|v}}, \tag{35}$$

$$(3), \ (4) \quad v_{max} \geq v > \frac{-Y_{22}(r) + \sqrt{\Delta_2}}{2Y_{|v|v}}, \tag{36}$$

where $\Delta_1 = Y_{22}^2(r) - 4Y_{|v|v}A$ and $\Delta_2 = Y_{22}^2(r) + 4Y_{|v|v}A$. It follows from this condition that $v$ is decreasing and therefore bounded. Considering that $\zeta_2 = v - \hat{Y}_{23}r$, it can be concluded that also $\zeta_2$ is limited.

Because the desired trajectory $u_d$ arises from trajectory planning and values of the velocity $u$ are limited by thrusters, then the desired trajectory is bounded. Moreover, the

error $y_e$ is bounded. From the analysis it follows that $v$ and $y_e$ are both bounded. Taking into account Young's inequality, i.e., $ab \leq \frac{\epsilon^2}{2}|a|^2 + \frac{1}{2\epsilon^2}|b|^2$ for $(a,b) \in \mathcal{R}$, where $\epsilon$ is a positive constant value [18], the following can be written:

$$
\begin{aligned}
\dot{V}_k &= -k_x x_e^2 - k_\psi \sin^2 \psi_e + v y_e \leq -k_x x_e^2 - k_\psi \sin^2 \psi_e + \frac{\epsilon^2}{2}|v|^2 + \frac{1}{2\epsilon^2}|y_e|^2 \\
&= -(k_x x_e^2 + k_\psi \sin^2 \psi_e - \frac{\epsilon^2}{2}|v|^2 - \frac{1}{2\epsilon^2}|y_e|^2).
\end{aligned}
\tag{37}
$$

From this it follows that $\dot{V}_k \leq 0$ as long as $k_x x_e^2 + k_\psi \sin^2 \psi_e > \rho_k$, where $\rho_k = \frac{\epsilon^2}{2}|v|^2 + \frac{1}{2\epsilon^2}|y_e|^2$. The symbol $\rho_k$ denotes a variable parameter that depends on the velocity $v$, the error $y_e$ and on the assumed value of $\epsilon$. Thus, selecting sufficiently great values of $k_x$, $k_\psi$ and appropriate value $\epsilon$ the velocity control subsystem will be asymptotically stable. However, if this condition is not met, then:

$$
\dot{V}_k \leq -(k_x x_e^2 + k_\psi \sin^2 \psi_e) + \rho_k.
\tag{38}
$$

This result means that all the time-varying signals are uniformly ultimately bounded (UUB) and the errors converge to a small neighborhood of zero. □

### 4.3. Dynamic Control Algorithm

The task of the dynamic controller is to move the vehicle to the desired velocities using $\tau_u$ and $\tau_r$, while the velocity errors are assumed as:

$$
[u_e, r_e]^T = [u - u_d, r - r_d]^T.
\tag{39}
$$

Because, the algorithm is expressed using IQV therefore $\zeta_{1e} = \zeta_1 - \zeta_{1d}$ and $\zeta_{3e} = \zeta_3 - \zeta_{3d}$ are applied. From (7)–(9) it is noticeable that it is also $u_e = u - u_d$ and $r_e = r - r_d$. For the surge force $\tau_u$ the following integral sliding surface is defined:

$$
S_1 = \zeta_{1e} + \lambda_1 \int_0^t \zeta_{1e}(\sigma)d\sigma,
\tag{40}
$$

where $\lambda_1 > 0$ is a control coefficient. The time derivative of $S_1$ taking into account (10) is $\dot{S}_1 = \dot{\zeta}_{1e} + \lambda_1 \zeta_{1e}$ (where $\dot{\zeta}_{1e} = \dot{\zeta}_1 - \dot{\zeta}_{1d}$):

$$
\dot{S}_1 = N_1^{-1}\left(F_1(\zeta) - N_1\dot{\zeta}_{1d} + N_1\lambda_1\zeta_{1e} + \tau_u + f_u\right).
\tag{41}
$$

To ensure the convergence of the error to zero along the sliding surface $S_1$ the following control algorithm is proposed:

$$
\tau_u = -\hat{F}_1(\zeta) + \hat{N}_1\dot{\zeta}_{1d} - \hat{N}_1\lambda_1\zeta_{1e} - \hat{f}_u - \Gamma_1\mathrm{sgn}(S_1),
\tag{42}
$$

where the last term, designed according to Assumption 3, is related to the parameters' perturbation reducing, and $\hat{f}_u$ means the estimated value of the external disturbance force $f_u$.

In order to prove the stability of the vehicle under the parameters' perturbation and external disturbances, the following Lyapunov function candidate is assumed:

$$
V_1 = \frac{1}{2}N_1 S_1^2 + \frac{1}{2}\beta_1\tilde{f}_u^2,
\tag{43}
$$

where $\tilde{f}_u = f_u - \hat{f}_u$ denotes the estimated error of the disturbance force and $\beta_1 > 0$ a constant coefficient (control parameter). The time derivative of $V_1$ calculated using (41) and (42) is:

$$\dot{V}_1 = N_1 S_1 \dot{S}_1 + \beta_1 \tilde{f}_u \dot{\tilde{f}}_u = S_1\big(F_1(\zeta) - \hat{F}_1(\zeta) + (\hat{N}_1 - N_1)\dot{\zeta}_{1d} + \lambda_1(N_1 - \hat{N}_1)\zeta_{1e}$$
$$-\Gamma_1 \mathrm{sgn}(S_1)\big) + \tilde{f}_u S_1 + \beta_1 \tilde{f}_u \dot{\tilde{f}}_u, \tag{44}$$

where:

$$F_1(\zeta) - \hat{F}_1(\zeta) = (m_{22} - \hat{m}_{22})vr + (m_{23} - \hat{m}_{23})r^2 + (\hat{d}_{11} - d_{11})u. \tag{45}$$

Next, making use of Assumption 1 the gain term $\Gamma_1$ is applied:

$$\Gamma_1 = \tilde{m}_{22}|vr| + \tilde{m}_{23}|r^2| + \tilde{d}_{11}|u| + \tilde{N}_1|\dot{\zeta}_{1d}| + \lambda_1 \tilde{N}_1|\zeta_{1e}| + \delta_1. \tag{46}$$

Then under Assumption 2 (if $\dot{f}_u = 0$, i.e., for constant disturbances), Equation (44) becomes:

$$\dot{V}_1 = N_1 S_1 \dot{S}_1 + \beta_1 \tilde{f}_u \dot{\tilde{f}}_u \leq -\delta_1|S_1| + \tilde{f}_u S_1 + \beta_1 \tilde{f}_u \dot{\tilde{f}}_u = -\delta_1|S_1| + \tilde{f}_u S_1 + \beta_1 \tilde{f}_u(\dot{f}_u - \dot{\hat{f}}_u)$$
$$\leq -\delta_1|S_1| + \tilde{f}_u(S_1 - \beta_1 \dot{\hat{f}}_u). \tag{47}$$

Designing the adaptive term $\dot{\hat{f}}_u = \beta_1^{-1} S_1$ one obtains:

$$\dot{V}_1 \leq -\delta_1|S_1| \leq 0. \tag{48}$$

This result means that $\dot{V}_1 = 0$ provided $\zeta_{1e}$ converges to zero along the sliding surface $S_1$. However, $\zeta_{1e} = u_e$, thus also $u_e$ tends to zero. Therefore, the surge velocity error $u_e$ converges asymptotically to zero if the signal $\tau_u$ (42) is used.

For the yaw moment (torque) the integral sliding surface is given in the form:

$$S_2 = \zeta_{3e} + \lambda_2 \int_0^t \zeta_{3e}(\sigma)d\sigma, \tag{49}$$

where $\lambda_2 > 0$ is a control coefficient. The time derivative of $S_2$ taking into account (12) is $\dot{S}_2 = \dot{\zeta}_{3e} + \lambda_2 \zeta_{3e}$ (where $\dot{\zeta}_{3e} = \dot{\zeta}_3 - \dot{\zeta}_{3d}$):

$$\dot{S}_2 = N_3^{-1}\big(F_3(\zeta) - N_3 \dot{\zeta}_{3d} + N_3 \lambda_2 \zeta_{3e} + \tau_r + f_{\zeta_3}\big). \tag{50}$$

The proposed input control signal $\tau_r$ has the form:

$$\tau_r = -\hat{F}_3(\zeta) + \hat{N}_3 \dot{\zeta}_{3d} - \hat{N}_3 \lambda_2 \zeta_{3e} - \hat{f}_{\zeta_3} - \Gamma_2 \mathrm{sgn}(S_2), \tag{51}$$

where the last term meets Assumption 3 and is related to the parameters' perturbation function, whereas $\hat{f}_{\zeta_3}$ is the estimated value of the external disturbance force $f_{\zeta_3}$.

The Lyapunov function candidate is assumed as follows:

$$V_2 = \frac{1}{2} N_3 S_2^2 + \frac{1}{2} \beta_2 \tilde{f}_{\zeta_3}^2, \tag{52}$$

where $\tilde{f}_{\zeta_3} = f_{\zeta_3} - \hat{f}_{\zeta_3}$ is the estimated error of the disturbance force and $\beta_2 > 0$ is a constant control parameter. Calculating the time derivative of $V_2$ and applying (50) and (51) it can be written:

$$\dot{V}_2 = N_3 S_2 \dot{S}_2 + \beta_2 \tilde{f}_{\zeta_3} \dot{\tilde{f}}_{\zeta_3} = S_2\big(F_3(\zeta) - \hat{F}_3(\zeta) + (\hat{N}_3 - N_3)\dot{\zeta}_{3d} + \lambda_2(N_3 - \hat{N}_3)\zeta_{3e}$$
$$-\Gamma_2 \mathrm{sgn}(S_2)\big) + \tilde{f}_{\zeta_3} S_2 + \beta_2 \tilde{f}_{\zeta_3} \dot{\tilde{f}}_{\zeta_3}, \tag{53}$$

where:

$$F_3(\zeta) - \hat{F}_3(\zeta) = (\hat{m}_{22} - m_{22})uv + (m_{11} - \hat{m}_{11})uv + (\hat{m}_{23} - m_{23})ur + \hat{Y}_{23}(\hat{m}_{11} - m_{11})ur$$
$$+ \hat{Y}_{23}(\hat{d}_{22} - d_{22})v + \hat{Y}_{23}(\hat{d}_{23} - d_{23})r + (\hat{d}_{32} - d_{32})v + (\hat{d}_{33} - d_{33})r. \tag{54}$$

Now using Assumption 1, the following gain function is adopted to guarantee the stability of the control system against parameter perturbations:

$$\Gamma_2 = (\widetilde{m}_{22} - \widetilde{m}_{11})|uv| + (\widetilde{m}_{23} + \hat{Y}_{23}\widetilde{m}_{11})|ur| + (\widetilde{d}_{32} + \hat{Y}_{23}\widetilde{d}_{22})|v|$$
$$+ (\widetilde{d}_{33} + \hat{Y}_{23}\widetilde{d}_{23})|r| + \widetilde{N}_3|\dot{\zeta}_{3d}| + \lambda_2\widetilde{N}_2|\zeta_{3e}| + \delta_2. \tag{55}$$

Then, given Assumption 2 and the first case, i.e., $\dot{f}_{\zeta_3} = 0$ (which implies from (12) that both $\dot{f}_v = 0$ and $\dot{f}_r = 0$, i.e., for constant disturbances), the function $\dot{V}_2$ (53) can be expressed as:

$$\dot{V}_2 = N_3 S_2 \dot{S}_2 + \beta_2 \tilde{f}_{\zeta_3} \dot{\tilde{f}}_{\zeta_3} \leq -\delta_2|S_2| + \tilde{f}_{\zeta_3} S_2 + \beta_2 \tilde{f}_{\zeta_3}(\dot{f}_{\zeta_3} - \dot{\hat{f}}_{\zeta_3})$$
$$\leq -\delta_2|S_2| + \tilde{f}_{\zeta_3}(S_2 - \beta_2 \dot{\hat{f}}_{\zeta_3}). \tag{56}$$

Designing the adaptive term $\dot{\hat{f}}_{\zeta_3} = \beta_2^{-1} S_2$ one obtains:

$$\dot{V}_2 \leq -\delta_2|S_2| \leq 0. \tag{57}$$

This result means that $\dot{V}_2 = 0$ provided $\zeta_{3e}$ converges to zero along the sliding surface $S_2$. However, $\zeta_{3e} = r_e$, thus also $r_e$ tends to zero. Therefore the yaw velocity error $r_e$ converges asymptotically to zero if the signal $\tau_r$ (51) is used.

Finally, for the entire control system, the following inequality is obtained:

$$\dot{V}_d = \dot{V}_1 + \dot{V}_2 \leq -\delta_1|S_1| - \delta_2|S_2| \leq 0, \tag{58}$$

which means that the proposed dynamic control algorithm can stabilize velocity errors $u_e$ and $r_e$.

**Remark 2.** *The discontinuous sign function can be replaced by using another smooth function which allows us to avoid the well-known chattering problem. For example, in [18], a definition of a smooth function was given. The properties of such a function satisfies, e.g., $f_s(\chi) = tanh(\chi)$. Replacing the discontinuous function sgn by the $tanh(a\chi)$ function, which, however, is only an approximation $sgn(\chi) \approx tanh(a\chi)$ where $a$ is a positive scalar can be found, e.g., in [9]. In [56] it was shown that if instead of using $sgn(\chi)$, the approximation by the use of $sat(\chi/\epsilon)$ is carried out, then the sliding mode control algorithm guarantees ultimate boundedness with an bound that can be controlled by the design of parameters $\epsilon$ (globally uniformly asymptotically stabilization can be proven). Taking the above into account and knowing that both functions, i.e., sat and tanh, are smooth saturation functions, one may conclude that also for the tanh function only globally uniformly asymptotically stabilization is ensured.*

**Remark 3.** *A more realistic situation than before will now be considered. This problem was discussed for manipulators in [57] and for underwater vehicles in [53]. If $\dot{f}_u \approx 0$ or $\dot{f}_u \neq 0$ the uncertainties are arbitrarily large and fast time-varying and then it is:*

$$\dot{V}_1 = N_1 S_1 \dot{S}_1 + \beta_1 \tilde{f}_u \dot{\tilde{f}}_u \leq -\delta_1|S_1| + \beta_1 \tilde{f}_u \dot{f}_u \leq -\delta_1|S_1| + \rho_1 \leq 0, \tag{59}$$

*where a positive scalar $\rho_1 \geq |\beta_1 \tilde{f}_u \dot{f}_u|$ can be found for the well-designed controller and bounded signal $\dot{f}_u$ (Assumption 2). It is necessary to meet the condition $|\beta_1 \tilde{f}_u \dot{f}_u| < \delta_1|S_1|$. However, in this case only uniformly ultimate boundedness will be guaranteed.*

If the uncertainties $\dot{f}_{\zeta_3} \approx 0$ or $\dot{f}_{\zeta_3} \neq 0$ (what means at least one of $\dot{f}_v$ and $\dot{f}_v$ is not equal to zero) then it is:

$$\dot{V}_2 = N_3 S_2 \dot{S}_2 + \beta_2 \tilde{f}_{\zeta_3} \dot{\hat{f}}_{\zeta_3} \leq -\delta_2 |S_2| + \beta_2 \tilde{f}_{\zeta_3} \dot{f}_{\zeta_3} \leq -\delta_2 |S_2| + \rho_2 \leq 0, \tag{60}$$

where a positive scalar $\rho_2 \geq |\beta_2 \tilde{f}_{\zeta_3} \dot{\hat{f}}_{\zeta_3}|$ can be found for the well-designed controller and bounded signal $\dot{f}_r$ (Assumption 2). It is necessary to meet the condition $|\beta_2 \tilde{f}_{\zeta_3} \dot{\hat{f}}_{\zeta_3}| < \delta_2 |S_2|$. However, in this case only uniformly ultimate boundedness will be guaranteed.

If the uncertainties $\dot{f}_u \approx 0$ or $\dot{f}_v \neq 0$, $\dot{f}_v \approx 0$ or $\dot{f}_u \neq 0$, $\dot{f}_r \approx 0$ or $\dot{f}_r \neq 0$ then one has:

$$\dot{V}_d = \dot{V}_1 + \dot{V}_2 \leq -\delta_1 |S_1| - \delta_2 |S_2| + \rho_3 \leq 0, \tag{61}$$

where a positive scalar $\rho_3 = \rho_1 + \rho_2$. In this case only uniformly ultimate boundedness will be guaranteed.

*4.4. Controller*

The obtained results can be summarized in the following proposition.

**Proposition 1.** *Consider an underactuated vehicle described by Equations (1) and (4)–(12) where the control objective is to stabilize trajectory tracking errors, i.e., $\lim_{t \to \infty} x_e = 0$, $\lim_{t \to \infty} y_e = 0$, and $\lim_{t \to \infty} \psi_e = 0$. If Assumptions 1–3 are fulfilled then the output signals $\tau_u$ (42) and $\tau_r$ (51) of the vehicle controller enable it to follow a desired trajectory in the horizontal plane in the presence of parameter perturbations and external disturbances. The full control scheme consists of:*

*(a) Kinematic control algorithm which makes the velocity control subsystem uniformly ultimately bounded;*

*(b) Dynamic control algorithm which moves the vehicle to a desired trajectory (and at least ensures uniformly ultimate boundedness).*

**Proof of Proposition 1.** Taking into considerations presented in Section 4.1 (assumptions), Sections 4.2 and 4.3 it can be concluded using the kinematic controller the velocity control subsystem is uniformly ultimately bounded, while the dynamic controller moves the vehicle to a desired trajectory and ensures uniformly ultimate boundedness. The control signals $\tau_u$ (42) and $\tau_r$ (51) guarantee that a desired trajectory in the horizontal plane under parameter perturbations and external disturbances is tracked. Thus, it can be stated that the stability analysis is completed. □

## 5. Simulations and Comparison

The thruster saturation effect is sometimes taken into consideration in the control algorithm [58]. However, when the thrust saturation effect is not serious, then the problem is omitted [8,52]. Such a situation occurs when the trajectories set for tracking are not complicated and the saturation effect for thrusters occurs only temporarily or occasionally.

*5.1. Vehicle Models and Test Conditions*

Simulation studies were performed to include at least partially real conditions. Two underwater vehicle models known from the literature were selected. Moreover, the technical capabilities resulting from the design of each vehicle were taken into account, namely the values of forces and moments which can be obtained from the thrusters and the possible vehicle velocities. The parameters of SIRENE [59] and Kambara [60] are given in Table 1. Since the test of the control algorithm was performed assuming the inertia matrix as symmetric, additional parameters were assumed in the simulations, namely: $m_{23} = m_{32} = -700$ kgm (for SIRENE), $m_{23} = m_{32} = -35$ kgm (for Kambara) and values $Y_r = N_v = 10$, $Y_{r|v|} = Y_{v|r|} = Y_{r|r|} = N_{r|v|} = N_{v|r|} = N_{v|v|} = 10$ (the same for both vehicles). It is worth noting that the parameters of the models of the two vehicles are significantly different, which allows

verification of the proposed algorithm for vehicles with different dynamic parameters. For tracking, the desired trajectory position profiles are assumed, described as $p_d = [x_r, y_r]^T$:

$$p_{d1} = [0.4\,t,\ 0.2\,t]^T, \tag{62}$$

$$p_{d2} = [0.6\,t - 3\cos(0.02\,t),\ 10 - 3\sin(0.02\,t)]^T, \tag{63}$$

linear and cycloid trajectory, respectively. The starting points (the same for both vehicles) were $p_{01} = [-5,\ 2,\ 0]^T$ and $p_{02} = [-5,\ 12,\ 0]^T$. These points were assumed in order to ensure realistic condition of work.

Moreover, time of motion was $t = 200$ s (for linear trajectory), $t = 500$ s (for cycloid trajectory) with the time step $\Delta t = 0.03$ s, and using the method ode 3 Bogacki–Shampine (in Matlab/Simulink). The simulation test was done based on software given in [61] but adapted for use with IQV.

Two types of disturbances were considered, i.e., constant and variable of the form (as an example of Assumption 2):

$$f_u = 12\text{ N},\ \ f_v = -1.5\text{ N},\ \ f_r = 2\text{ Nm}, \tag{64}$$

$$f_u(t) = 2 + 1.5\sin(0.3\,t) + 0.5\cos(0.2\,t)\text{ N}, \tag{65}$$

$$f_v(t) = 1 + 0.5\sin(0.1\,t) + 0.3\cos(0.3\,t)\text{ N}, \tag{66}$$

$$f_r(t) = 1 + \sin(0.2\,t) + 0.2\cos(0.4\,t)\text{ Nm}, \tag{67}$$

**Table 1.** Parameters of SIRENE and Kambara.

|  | SIRENE | Kambara |  |
| --- | --- | --- | --- |
| **Symbol** | **Value** | **Value** | **Unit** |
| $L$ | 4.0 | 1.2 | m |
| $b$ | 1.6 | 1.5 | m |
| $h$ | 1.96 | 0.9 | m |
| $m_{11}$ | 2234.5 | 175.4 | kg |
| $m_{22}$ | 2234.5 | 140.8 | kg |
| $m_{33}$ | 2000 | 16.07 | $\text{kgm}^2$ |
| $X_u$ | 0 | 120 | kg/s |
| $Y_v$ | 346 | 90 | kg/s |
| $N_r$ | 1427.2 | 18 | $\text{kgm}^2/\text{s}$ |
| $X_{u|u|}$ | 35.4090 | 90 | kg/m |
| $Y_{v|v|}$ | 667.5552 | 90 | kg/m |
| $N_{r|r|}$ | 26,036 | 15 | $\text{kgm}^2$ |

For evaluating the tracking control performance the following indexes were assumed—*MIA* (mean integrated absolute error), *MIAC* (mean integrated absolute control), *RMS* (root mean square of the tracking error), i.e.,

$$MIA = \frac{1}{t_f - t_0}\int_{t_0}^{t_f} |\pi_e(t)|dt, \quad MIAC = \frac{1}{t_f - t_0}\int_{t_0}^{t_f} |\tau(t)|dt,$$

$$RMS = \sqrt{\frac{1}{t_f - t_0}\int_{t_0}^{t_f} \|e(t)\|dt}, \quad K_m = \text{mean}(KE), \tag{68}$$

where $\pi_e = x_e, y_e, \psi_e, \ldots$, $\|e(t)\| = \sqrt{x_e^E + y_e^E}$ ($x_e^E$, $y_e^E$ mean the position error in the reference frame) and $KE$ is the kinetic energy, respectively.

A heuristic method was used to select control parameters, as in [51]. This method is similar to the trial-and-error method, but also reflects the dynamics of the vehicle model.

5.1.1. SIRENE Test Using Proposed Algorithm

The vehicle data were taken from [59,62]. Conditions of the investigation were as follows: the forces and torques values were limited here, based on the references, i.e., $|\tau_u| \leq 500$ N, $|\tau_r| \leq 500$ Nm, and the velocities $u_{max} = 2$ m/s, $v_{max} = 0.8$ m/s. The operating conditions of the vehicle are in accordance with the technical data provided in the cited literature.

For the controller used, the inaccuracy of the model parameters was assumed as W = 0.2 (20%) and the following set of control parameters:

$$k_x = 20, \quad k_\psi = 2, \quad \lambda_1 = \lambda_2 = 0.1, \quad \Gamma_1 = \Gamma_2 = 0.1, \quad \beta_1 = \beta_2 = 0.1. \qquad (69)$$

The set of coefficients was selected to ensure that the tracking task could be carried out under the specified constraints.

The results for the desired linear trajectory (62) and constant disturbances (64) are given in Figure 2. From Figure 2a–c it can be seen that the trajectory is tracked correctly. Likewise, the velocities $u, r$ tend towards the desired ones (large values of $u_d$ and $r_d$ at the beginning of the movement are due to the conditions of the method and the vehicle dynamics) as it arises from Figure 2d,e. All velocities $u, v, r$ have acceptable values (Figure 2f) and force $\tau_u$ and torque $\tau_r$ only have large values at the beginning, as results from Figure 2g.

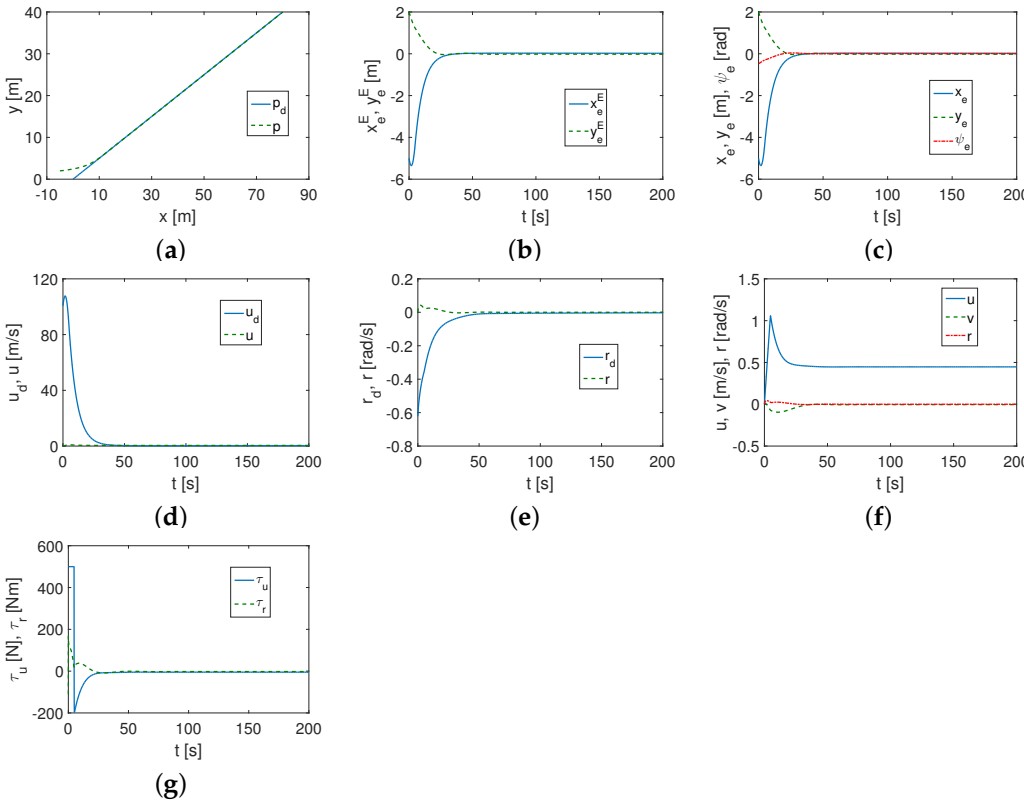

**Figure 2.** Simulation results for SIRENE–IQV controller and linear trajectory (constant disturbances): (**a**) desired and realized trajectory; (**b**) position errors (E-frame); (**c**) position errors (B-frame) and angular error; (**d**) desired and realized velocity ($u_d$, $u$); (**e**) desired and realized velocity ($r_d$, $r$); (**f**) realized velocities; (**g**) applied force and torque.

Very similar results were obtained for the same trajectory but using variable disturbances (65)–(67). They are shown in Figure 3a–g.

Simulation results for the cycloid trajectory (63) and constant disturbances (64) are presented in Figure 4a–g. As can be seen, also in the case of a curvilinear trajectory, the tracking task is carried out correctly. Compared to the previous graphs, some fluctuations of the variables can be observed (position, velocity, control signals).

When the variable disturbances described by (65)–(67) were applied, one can found that the performance was very close to that obtained for the constant disturbances as can be observed from Figure 5a–g.

For a more accurate comparison, the results obtained using criteria (68) are summarized in Table 2 (C. dist.—constant disturbances, V. dist.—variable disturbances).

It was found that for the SIRENE vehicle, the mean position and velocity errors were smaller when a cycloid trajectory was used instead of the linear trajectory. The forces and the torque also had smaller values while the mean kinetic energy had a significantly larger value. The use of constant and variable disturbances (with the given parameters) resulted in only minor changes in the performance.

**Table 2.** Performance for SIRENE.

| | | Linear | Trajectory | Cycloid | Trajectory |
|---|---|---|---|---|---|
| **Index** | | **C. Dist.** | **V. Dist.** | **C. Dist.** | **V. Dist.** |
| MIA | $x_e^E$ | 0.2807 | 0.2821 | 0.0752 | 0.0757 |
| | $y_e^E$ | 0.0972 | 0.1047 | 0.0771 | 0.0756 |
| | $\psi_e$ | 0.0287 | 0.0262 | 0.0155 | 0.0128 |
| | $x_e$ | 0.2807 | 0.2820 | 0.0753 | 0.0759 |
| | $y_e$ | 0.0981 | 0.1054 | 0.0769 | 0.0754 |
| | $u_e$ | 5.3909 | 9.4245 | 1.3491 | 1.3603 |
| | $r_e$ | 0.0360 | 0.0353 | 0.0361 | 0.0362 |
| MIAC | $\tau_u$ | 22.817 | 21.654 | 7.9840 | 15.953 |
| | $\tau_r$ | 6.2126 | 5.2778 | 4.5685 | 4.2121 |
| RMS | $\|e\|$ | 1.0112 | 1.0176 | 0.4257 | 0.4297 |
| KE | $K_m$ | 253.92 | 253.96 | 429.61 | 429.61 |

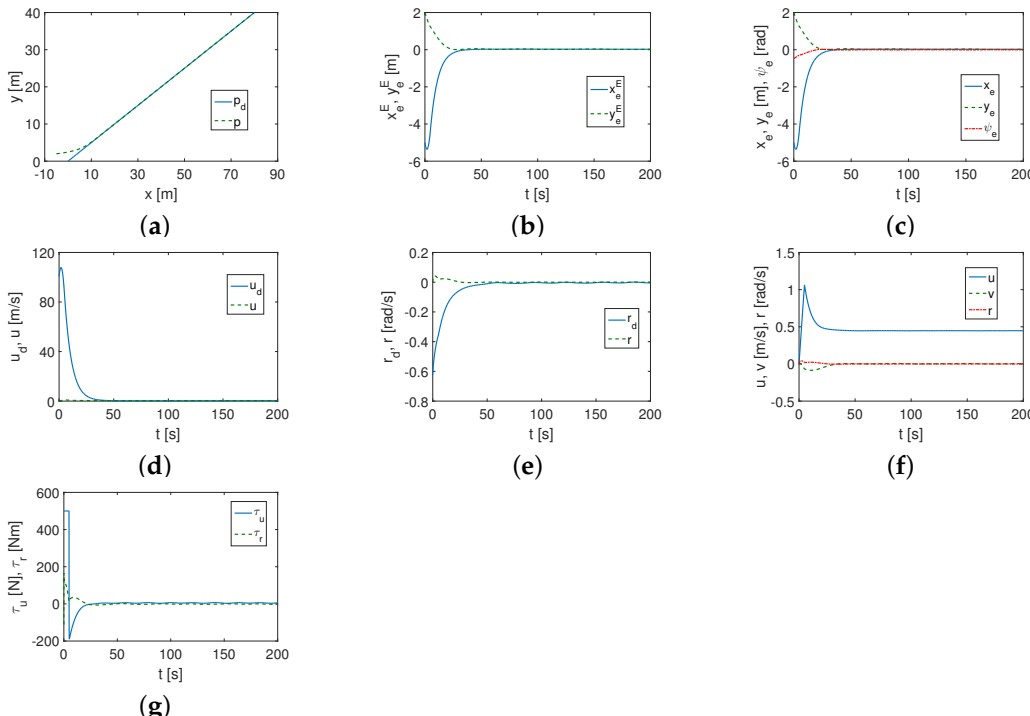

**Figure 3.** Simulation results for SIRENE—IQV controller and linear trajectory (variable disturbances): (**a**) desired and realized trajectory; (**b**) position errors (E-frame); (**c**) position errors (B-frame) and angular error; (**d**) desired and realized velocity ($u_d$, $u$); (**e**) desired and realized velocity ($r_d$, $r$); (**f**) realized velocities; (**g**) applied force and torque.

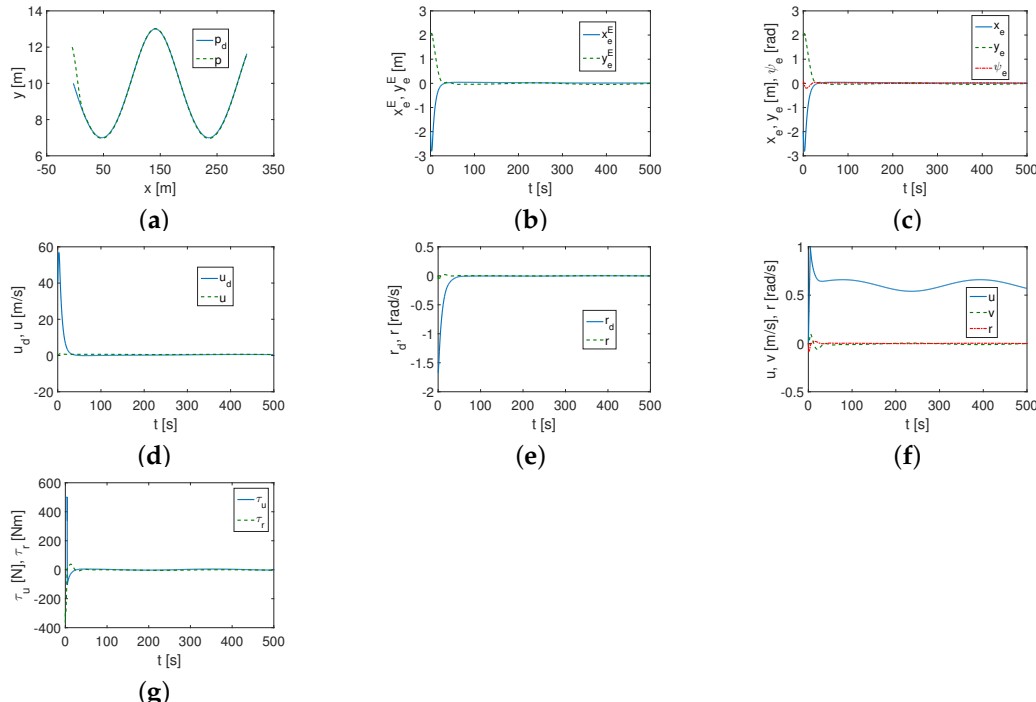

**Figure 4.** Simulation results for SIRENE—IQV controller and cycloid trajectory (constant disturbances): (**a**) desired and realized trajectory; (**b**) position errors (E-frame); (**c**) position errors (B-frame) and angular error; (**d**) desired and realized velocity ($u_d$, $u$); (**e**) desired and realized velocity ($r_d$, $r$); (**f**) realized velocities; (**g**) applied force and torque.

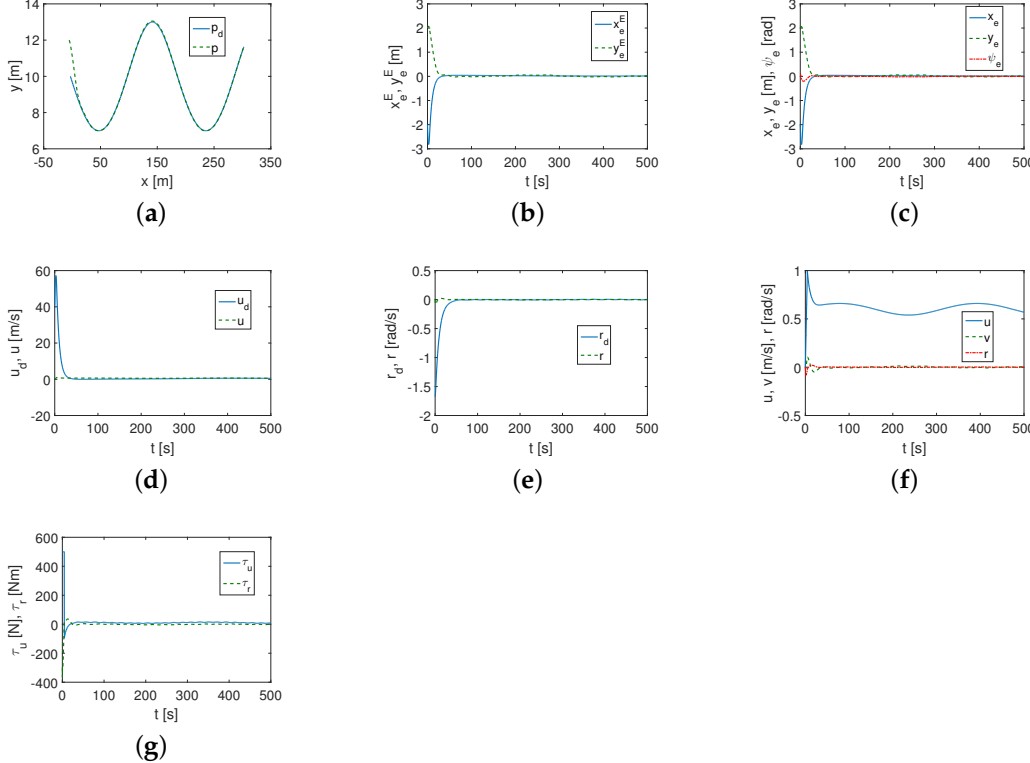

**Figure 5.** Simulation results for SIRENE—IQV controller and cycloid trajectory (variable disturbances): (**a**) desired and realized trajectory; (**b**) position errors (E-frame); (**c**) position errors (B-frame) and angular error; (**d**) desired and realized velocity ($u_d$, $u$); (**e**) desired and realized velocity ($r_d$, $r$); (**f**) realized velocities; (**g**) applied force and torque.

### 5.1.2. Kambara Test Using Proposed Algorithm

The vehicle data come from [60,63]. Conditions of the investigation were as follows: the forces and torques values were limited here, based on references, i.e., $|\tau_u| \leq 140$ N, $|\tau_r| \leq 8$ Nm, and the velocities $u_{max} = 1$ m/s, $v_{max} = 0.2$ m/s.

For the controller used, the inaccuracy of the model parameters W = 0.2 (20%) and the following set of control parameters were assumed:

$$k_x = 25, \quad k_\psi = 1.2, \quad \lambda_1 = \lambda_2 = 0.1, \quad \Gamma_1 = \Gamma_2 = 0.1, \quad \beta_1 = 0.2, \quad \beta_2 = 0.01. \quad (70)$$

The set of coefficients was chosen to ensure that the tracking task is accomplished under the constraints introduced.

Figure 6 shows the results obtained for linear trajectory (62) tracking and constant disturbances (64). From Figure 6a–e it can be observed that the position tracking task is realized correctly and the velocities $u, r$ tend towards the desired ones. The velocities $u, v, r$ are below the limits (Figure 6f) as well as the force $\tau_u$ and torque $\tau_r$ are limited with exception the first phase of motion (Figure 6g).

Using time-varying disturbances (65)–(67), the controller also accomplished the trajectory tracking task as shown in Figure 7a–g, but the effect of these disturbances is clearly visible for position, velocity, and force and moment errors (fluctuating variables).

Next, in Figure 8a–g, simulation results for the cycloid trajectory (63) and constant disturbances (64) are given. These are similar to those obtained for the linear trajectory and perturbation constants shown in Figure 6a–g. However, such movement causes some signal changes during trajectory tracking.

For the variable disturbances (65)–(67), the performance shown in Figure 9a–g was very close to that obtained for the time-varying disturbances as can be noted from Figure 7a–g.

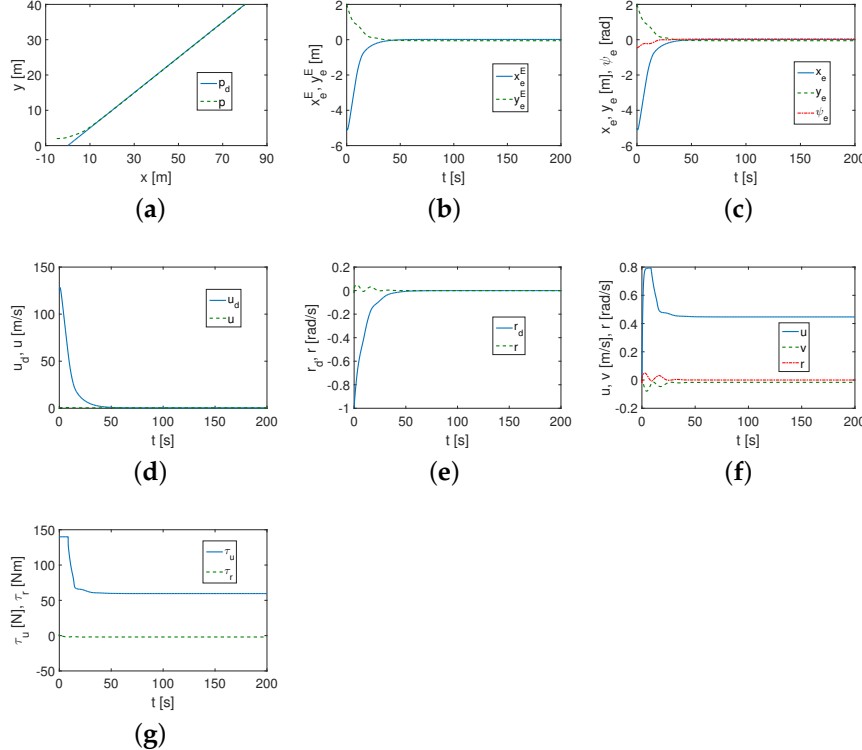

**Figure 6.** Simulation results for Kambara—IQV controller and linear trajectory (constant disturbances): (**a**) desired and realized trajectory; (**b**) position errors (E-frame); (**c**) position errors (B-frame) and angular error; (**d**) desired and realized velocity ($u_d, u$); (**e**) desired and realized velocity ($r_d, r$); (**f**) realized velocities; (**g**) applied force and torque.

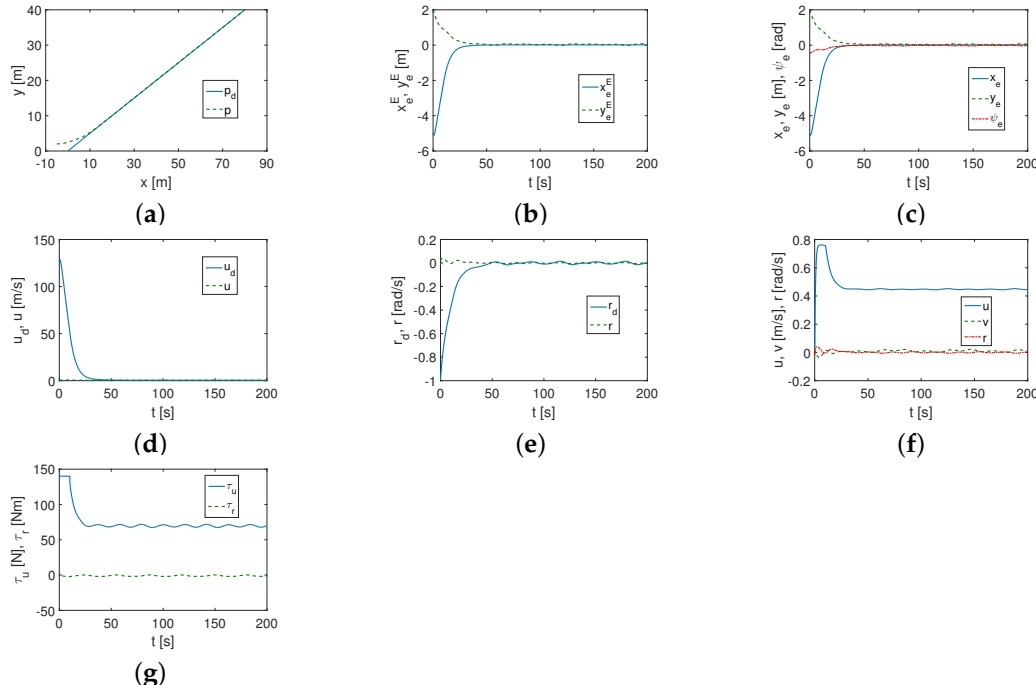

**Figure 7.** Simulation results for Kambara—IQV controller and linear trajectory (variable disturbances): (**a**) desired and realized trajectory; (**b**) position errors (E-frame); (**c**) position errors (B-frame) and angular error; (**d**) desired and realized velocity ($u_d$, $u$); (**e**) desired and realized velocity ($r_d$, $r$); (**f**) realized velocities; (**g**) applied force and torque.

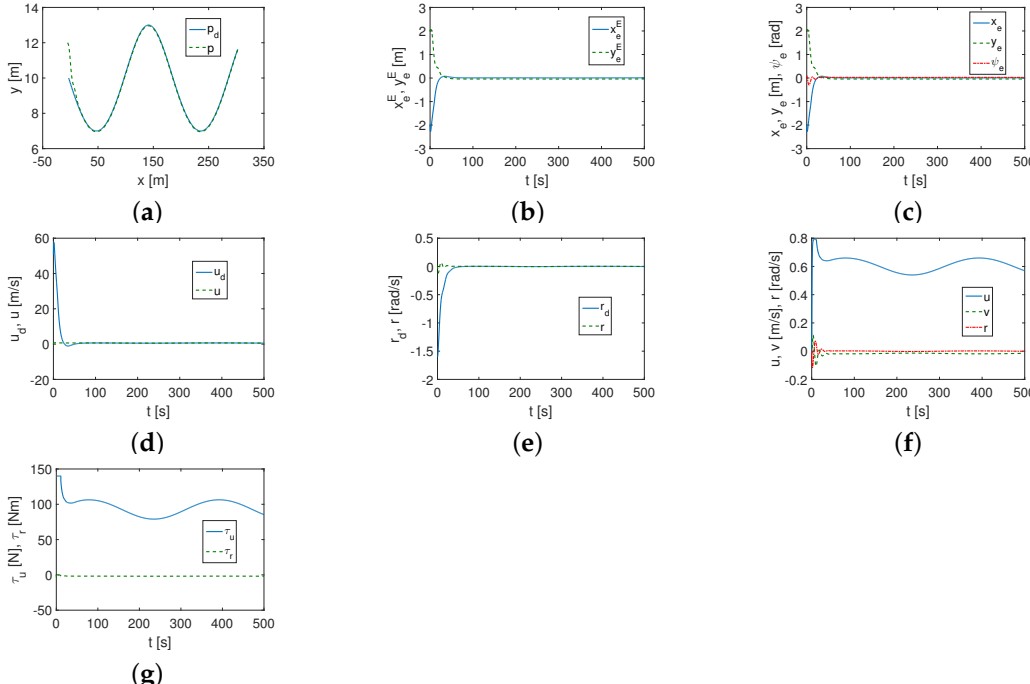

**Figure 8.** Simulation results for Kambara—IQV controller and cycloid trajectory (constant disturbances): (**a**) desired and realized trajectory; (**b**) position errors (E-frame); (**c**) position errors (B-frame) and angular error; (**d**) desired and realized velocity ($u_d$, $u$); (**e**) desired and realized velocity ($r_d$, $r$); (**f**) realized velocities; (**g**) applied force and torque.

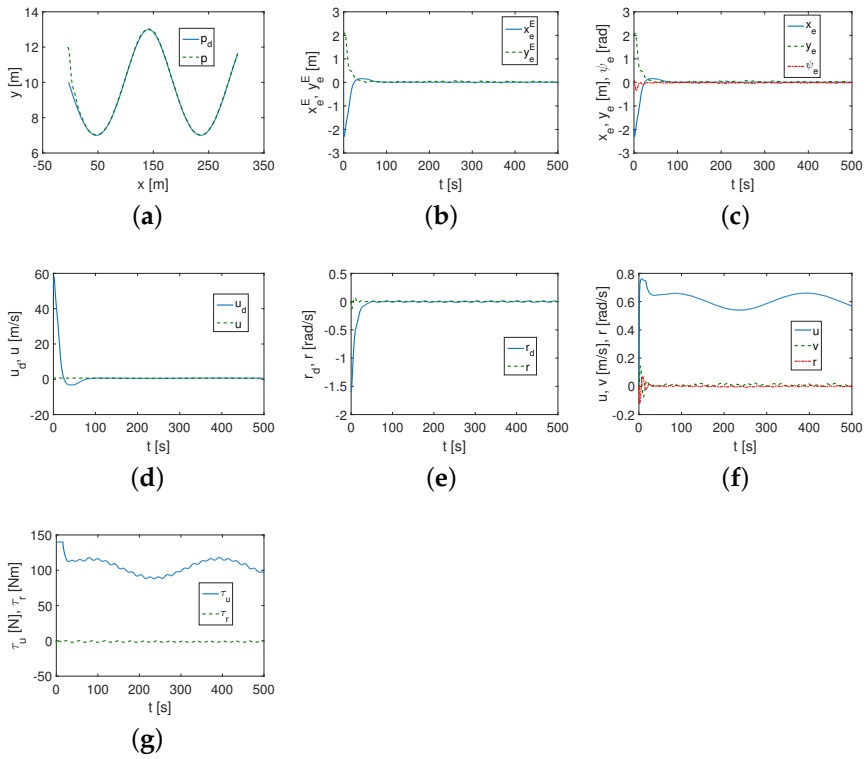

**Figure 9.** Simulation results for Kambara—IQV controller and cycloid trajectory (variable disturbances): (**a**) desired and realized trajectory; (**b**) position errors (E-frame); (**c**) position errors (B-frame) and angular error; (**d**) desired and realized velocity ($u_d$, $u$); (**e**) desired and realized velocity ($r_d$, $r$); (**f**) realized velocities; (**g**) applied force and torque.

For a more objective evaluation, criteria (68) were used and summarized in Table 3.

**Table 3.** Performance for Kambara.

|         |             | **Linear** | **Trajectory** | **Cycloid** | **Trajectory** |
|---------|-------------|:----------:|:--------------:|:-----------:|:--------------:|
| **Index** |           | **C. Dist.** | **V. Dist.** | **C. Dist.** | **V. Dist.** |
| MIA     | $x_e^E$     | 0.2573     | 0.2583         | 0.0530      | 0.0700         |
|         | $y_e^E$     | 0.1371     | 0.1425         | 0.0855      | 0.0793         |
|         | $\psi_e$    | 0.0518     | 0.0516         | 0.0302      | 0.0223         |
|         | $x_e$       | 0.2566     | 0.2582         | 0.0531      | 0.0702         |
|         | $y_e$       | 0.1380     | 0.1432         | 0.0853      | 0.0791         |
|         | $u_e$       | 6.2570     | 6.2901         | 1.1692      | 1.6046         |
|         | $r_e$       | 0.0505     | 0.0544         | 0.0321      | 0.0371         |
| MIAC    | $\tau_u$    | 64.652     | 74.632         | 96.729      | 106.75         |
|         | $\tau_r$    | 1.8703     | 1.1113         | 1.8144      | 1.1432         |
| RMS     | $\|\|e\|\|$ | 0.9303     | 0.9671         | 0.3643      | 0.3950         |
| KE      | $K_m$       | 19.679     | 19.662         | 33.637      | 33.635         |

It can be observed that for the Kambara vehicle, the mean position and velocity errors are smaller when a cycloid trajectory is used instead of the linear trajectory. The mean force $\tau_u$ and the mean kinetic energy have however greater values. It may be observed that the differences in system response in the presence of fixed and time-varying disturbances are small. However, as can be seen from the figures shown there are differences in the variables obtained (oscillation of some signals).

### 5.1.3. Comparison with Another Controller

The trajectory tracking algorithm proposed in [42] was chosen for comparative simulation studies. It is suitable for marine vehicles in horizontal motion, for which the inertia matrix in the mathematical model is symmetric. The reason for choosing this algorithm is that it, too, is based on a transformation of variables, except that instead of a velocity transformation (as in the proposed control scheme), it uses a transformation of coordinates describing the vehicle's position. Moreover, this control scheme has the advantage of having been tested in both simulation studies and a sea trial. In both tests it proved to be effective. In this regard, its effectiveness was compared with that of the proposed controller by simulation. This is not equivalent to an experiment, of course, but it indirectly provides information on the importance of the presented control approach.

Simulation tests were conducted according to the method described in detail in [42]. In this work, the tests performed for this controller are designated CL (classical controller). Verification of the selected algorithm could be performed only for constant velocity disturbances as these are the requirements of its applicability. The vehicles and trajectories used for tracking were as in the tests of the previously proposed control algorithm. The transformed equations of motion described in [42] are as follows ($z_1 = \psi, z_2 = r$):

$$\dot{z}_1 = z_2, \tag{71}$$

$$\dot{z}_2 = F_{z_2}(z_1, \xi_3, \xi_4) + \tau_r, \tag{72}$$

$$\begin{bmatrix} \dot{\xi}_1 \\ \dot{\xi}_2 \end{bmatrix} = \begin{bmatrix} \xi_1 \\ \xi_2 \end{bmatrix} + \begin{bmatrix} V_x \\ V_y \end{bmatrix}, \tag{73}$$

$$\begin{bmatrix} \dot{\xi}_3 \\ \dot{\xi}_4 \end{bmatrix} = \begin{bmatrix} F_{\xi_3}(z_1, \xi_3, \xi_4) \\ F_{\xi_4}(z_1, \xi_3, \xi_4) \end{bmatrix} + \begin{bmatrix} \cos z_1 & -l \sin z_1 \\ \sin z_1 & l \cos z_1 \end{bmatrix} \begin{bmatrix} \tau_u \\ \tau_r \end{bmatrix}. \tag{74}$$

The input signals are expressed as:

$$\begin{bmatrix} \tau_u \\ \tau_r \end{bmatrix} = \begin{bmatrix} \cos \psi & -l \sin \psi \\ \sin \psi & l \cos \psi \end{bmatrix}^{-1} \begin{bmatrix} -F_{\xi_3}(z_1, \xi_3, \xi_4) + \mu_1 \\ -F_{\xi_4}(z_1, \xi_3, \xi_4) + \mu_2 \end{bmatrix}, \tag{75}$$

with the control inputs:

$$\mu_1 = -k_{v_x}(\xi_3 - \xi_{3_d}) - k_{p_x}(\xi_1 - \xi_{1_d}) - k_{I_x}(\xi_{1_I} - \xi_{1_{dI}}) + \dot{\xi}_{3_d}, \tag{76}$$

$$\mu_2 = -k_{v_y}(\xi_4 - \xi_{4_d}) - k_{p_y}(\xi_2 - \xi_{2_d}) - k_{I_y}(\xi_{2_I} - \xi_{2_{dI}}) + \dot{\xi}_{4_d}, \tag{77}$$

where $k_{v_x}, k_{v_y}, k_{p_x}, k_{p_y}, k_{I_x}$, and $k_{I_y}$ mean positive real gains and $\xi_I$ where $i \in \{1, 2, 1_d, 2_d\}$ are integrals of the appropriate signals. All symbols are defined in the cited reference.

According to the requirements of the control method, the following parameters for SIRENE, velocity disturbances $V_x = 0.1$ m/s, $V_y = -0.1$ m/s, and linear trajectory were assumed:

$$k_{v_x} = 5.5, \quad k_{v_y} = 5.5, \quad k_{p_x} = 1.0, \quad k_{p_y} = 1.4, \quad k_{I_x} = 0.02, \quad k_{I_y} = 0.02, \tag{78}$$

and $l = 1.5$ (to avoid oscillations). As can be seen from Figure 10a, the desired trajectory is not tracked exactly, which is confirmed in Figure 10b; in spite of that, the quantities $\Delta\xi_1, \Delta\xi_2, \Delta\xi_3, \Delta\xi_4$ (Figure 10c,d) tend to zero which is a requirement for the effectiveness of the control algorithm. The velocities (Figure 10e) are permitted and the force and torque are small (Figure 10f). For the cycloid trajectory the algorithm failed.

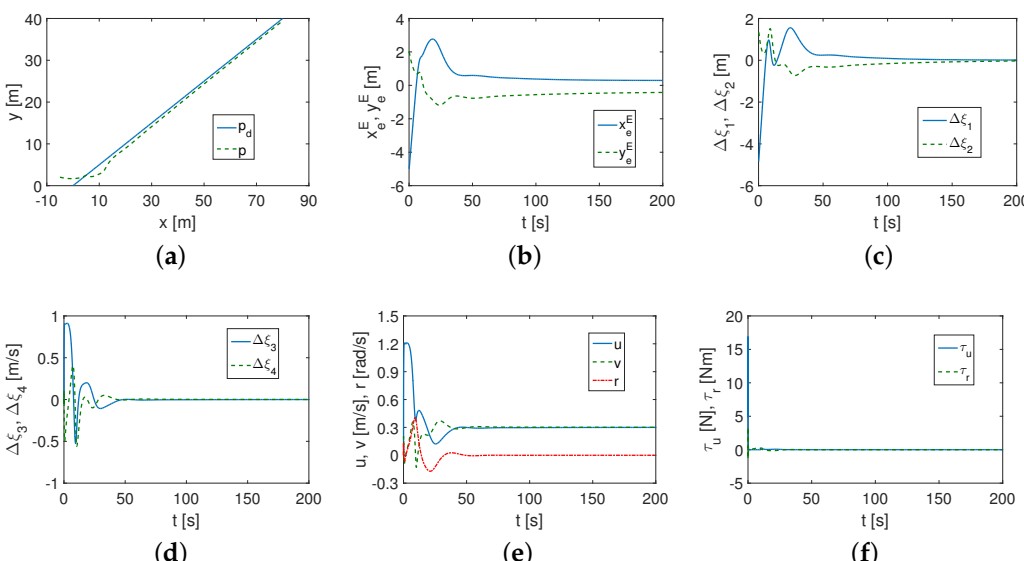

**Figure 10.** Simulation results for SIRENE, CL controller and linear trajectory: (**a**) desired and realized trajectory; (**b**) position errors; (**c**) position error states; (**d**) velocity error states; (**e**) velocities; (**f**) applied force and torque.

Next, the simulation test of the controller for Kambara, using the same velocity disturbances and for both linear and cycloid trajectory, was performed. The following set of parameters was applied:

$$k_{v_x} = 20, \quad k_{v_y} = 20, \quad k_{p_x} = 2.5, \quad k_{p_y} = 2.5, \quad k_{I_x} = 0.1, \quad k_{I_y} = 0.1, \qquad (79)$$

and $l = 1.2$ (to avoid oscillations). As it shown in Figure 11a, the desired linear trajectory is also not tracked exactly, which is confirmed in Figure 11b here it is seen that the position errors do not tend to zero. On the contrary, all variables $\Delta\xi_1, \Delta\xi_2, \Delta\xi_3, \Delta\xi_4$ go to zero in the considered time as is presented in Figure 11c,d. Moreover, velocity $v$ given in Figure 11e exceeds the allowed value assumed for the test, whereas the force and torque have small values (Figure 11f).

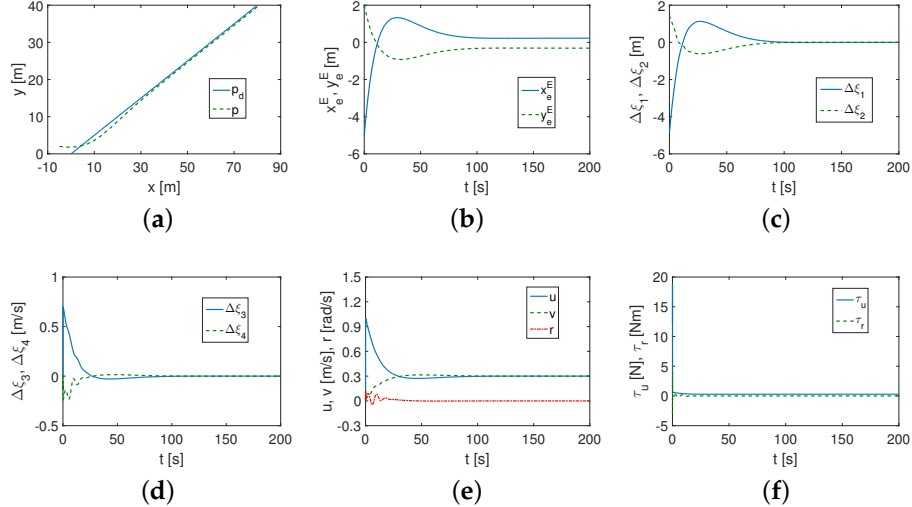

**Figure 11.** Simulation results for Kambara, CL controller and linear trajectory: (**a**) desired and realized trajectory; (**b**) position errors; (**c**) position error states; (**d**) velocity error states; (**e**) velocities; (**f**) applied force and torque.

For the cycloid trajectory, the results are shown in Figure 12. From Figure 12a,b, it can be observed that the trajectory position tracking errors are significant and thus the tracking task is not performed satisfactorily. Meanwhile, the time history of the variables $\Delta \xi_1, \Delta \xi_2, \Delta \xi_3, \Delta \xi_4$, as is shown in Figure 12c,d, indicates that they converge to zero. The velocity $r$ (Figure 12e) has big value (although this is an acceptable value). In practice, however, this means that the vehicle will perform an oscillating motion about a vertical axis. The applied force and torque have very small values (Figure 12f).

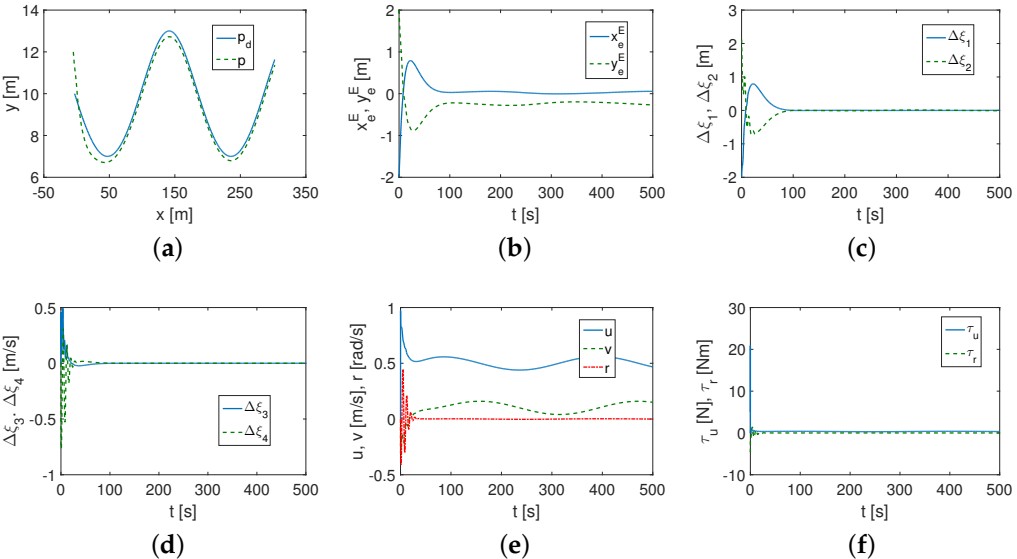

**Figure 12.** Simulation results for Kambara, CL controller and cycloid trajectory: (**a**) desired and realized trajectory; (**b**) position errors; (**c**) position error states; (**d**) velocity error states; (**e**) velocities; (**f**) applied force and torque.

The performance of the algorithm is collected in Table 4. For SIRENE values of indexes are grater than for the proposed control algorithm (with exception of MIAC). Note however that the mean kinetic energy is comparable for both controllers. Similar observation can be made for the Kambara vehicle (but the mean kinetic energy is slightly larger for the proposed algorithm).

**Table 4.** Performance for SIRENE and Kambara.

|  |  | SIRENE Linear t. | Kambara Linear t. | Kambara Cycloid t. |
|---|---|---|---|---|
| **Index** |  | **C. Dist.** | **C. Dist.** | **C. Dist.** |
| MIA | $x_e^E$ | 0.6669 | 0.5464 | 0.0953 |
|  | $y_e^E$ | 0.6032 | 0.4637 | 0.2962 |
|  | $\psi_e$ | 0.4111 | 0.3167 | 0.1988 |
| MIAC | $\tau_u$ | 0.0117 | 0.3051 | 0.3627 |
|  | $\tau_r$ | 0.0196 | 0.0117 | 0.0246 |
| RMS | $\|e\|$ | 1.2537 | 0.9804 | 0.4147 |
| KE | $K_m$ | 252.74 | 16.377 | 24.452 |

An additional test was done for velocity disturbances $V_x = 0.4\,\text{m/s}$, $V_y = -0.5\,\text{m/s}$. In this case, it was found that the results are much worse, which means significantly larger tracking errors (according to the assumed indexes) and a significant increase in average kinetic energy (mostly more than doubled, with the exception of the cycloid trajectory for Kambara).

5.1.4. Discussion of Results

In order to verify the effectiveness of the proposed trajectory tracking algorithm, simulations were carried out on models of two vehicles with significantly different dynamics. Two types of trajectories (linear and curvilinear) were also considered, as well as perturbation functions of constant and variable value.

For the developed algorithm, the values of the control parameters were only slightly different for the two vehicle models and the trajectories implemented. This may suggest that in the proposed control scheme, the dynamic parameters have a significant impact on vehicle motion and the controller gains are tuned to improve performance (for the controller used for comparison, the control parameters had to be significantly changed).

The indexes for control errors and had similar values for both vehicles but differed depending on the trajectory being tested (the effect of the disturbance function was less significant). Kinetic energy consumption depended on the vehicle and the trajectory realized. On the other hand, the figures show that the time to reach the desired trajectory is short considering the weight of the vehicles and the limitations of the thrusters (according to the technical data).

The comparative simulations show that the control algorithm described in [42] can work for various vehicles, but its effectiveness depends on the dynamic parameters of the vehicle, the trajectory realized and the operating conditions. Tracking time is very long and velocity disturbances must have small values (about 0.1 m/s). Compared to this controller, the proposed algorithm has a slightly larger range of applicability (the interference function can be varied) and the position errors are adjusted directly (without coordinate transformation), which makes it possible to reduce their values more effectively.

Taking into account the simulation studies conducted, the following conclusions can be made: (1) the control algorithm can be applied to different vehicles, but its effectiveness may vary due to different model parameters (mass, drag coefficients, etc.); (2) the results of trajectory tracking depend on the shape of the desired trajectory (linear, curvilinear); (3) external disturbances reduce the effectiveness of the algorithm, and this effectiveness depends on whether they have constant values or are changed during the movement of the vehicle; (4) the consumption of kinetic energy and the applied force and torque required to perform the task depend on the shape of the trajectory, and to a lesser extent on the realized trajectory (for small disturbances such as those used in this work).

**6. Comments on the Proposed Algorithm Using IQV**

The proposed control algorithm was developed for the diagonalized equations of dynamics, which were obtained from the decomposition of a symmetric inertia matrix. However, the algorithm in a simplified version is also suitable for trajectory tracking when the vehicle is described by a model with a diagonal inertia matrix.

In the control strategies of recent years, it is very common that the inertia matrix is diagonal. This means that dynamic couplings are compensated by the controller. Information about the couplings is then not available. To guarantee good performance of the controller, combinations of different methods are also used. For example, the use of the following control methods can be given:

1. SMC [9,10] or SMC and backstepping [16];
2. NN based controllers [26] (with SMC), [16,17,23,37] (with SMC and backstepping), [31] (and event-triggered control);
3. Observer-based control schemes [30] (event-triggered dynamic surface control), [28] (using fuzzy logic), [40] (line-of-sight (LOS) adaptive trajectory tracking controller with terminal sliding mode control);
4. Nonlinear model predictive control (NMPC) strategy [22];
5. Controllers with prescribed performance [33].

The proposed controller based on IQV uses SMC and backstepping methods, but also velocity transformation. Thus, it is a slightly different combination than the methods in Group 1. In NN-based methods (Group 2), additional knowledge (about NN) is required,

which can improve the performance of the controller, but at the same time the control scheme is more complicated. Control strategies based on observer (Group 2) allow the use of inaccessible signals. However, the dynamics of the system with an observer also changes, which distorts the information about the dynamics of the system itself (i.e., without an observer). The IQV controller only takes into account the vehicle model. In [22], additional optimization was used, and such a mechanism is not included in the developed IQV algorithm. In the prescribed performance based methods as in [33], it is necessary to guarantee that the signals run within certain limits, while the IQV controller does not have to meet this condition.

It is worth recalling some control strategies suitable for asymmetric vehicles because the proposed algorithm belongs to this group. However, one should point out the differences between it and selected controllers developed in recent years and performing the same task.

1. SMC [39];
2. Controllers using NN [34], ref. [36] (disturbance observer, backstepping, prescribed performance);
3. Observer-based approaches [41] (prescribed performance), ref. [44] (Lyapunov approach), ref. [43] (Lyapunov and backstepping control methods);
4. Input–output linearization [42];
5. Controllers with prescribed performance [64].

The differences between the proposed algorithm and the previously mentioned control methods have already been indicated. However, the methods for vehicle models with asymmetry serve the same purpose as the proposed algorithm. The method described in [42] uses the transformation of coordinates and for this reason it was chosen for comparison in the simulation test.

This article focuses on showing the operation of the control scheme using IQV, but ignores its other important property known in the literature. The dynamical equations denoted can also be used to study vehicle dynamics (e.g., coupling estimation), as shown for fully actuated vehicles in [49]. An identical possibility is not provided by the aforementioned control strategies for asymmetric vehicles, because, although the model equations can be transformed so that the accelerations are independent, e.g., in [36,42,64], the inertial couplings are only transformed (the diagonalizing method of the acceleration equations is not used). In addition, the authors of these papers did not indicate that their proposed strategies can be used to analyze vehicle dynamics, while the offered controller can be used for this purpose during the trajectory tracking task.

The algorithm developed here serves only for tracking desired trajectories, while it is not useful for path tracking. For this reason, the results obtained using it cannot be compared with the performance of the control schemes considered in [1–3]. In contrast, the control strategy proposed in [42] would be suitable for such comparisons because it is more universal than the algorithm presented in this paper and is suitable for following the path.

Thus, it is possible to point out some advantages that result from using the proposed algorithm with IQV:

- It can be applied to asymmetric vehicles, and thereby to a dynamic model more realistic than the model with a diagonal inertia matrix;
- Selection of controller parameters is intuitive and does not require any additional search methods (this can be explained by the fact that the dynamic parameters are already included in the algorithm, which makes it easier to select the controller parameters);
- In contrast to the usually used algorithms, it gives a potential possibility to estimate the effect of couplings on the vehicle behavior in motion (this is possible because after the velocity transformation one can obtain additional information hidden in the symmetric inertia matrix, but this issue, however, was not the subject of this paper);
- This is also an extension of the IQV control concept for underatuated vehicles with 3 DOF (since for fully activated marine vehicles there are algorithms for up to 6 DOF, such as, e.g., in [49]).

## 7. Conclusions

In this paper a trajectory tracking problem for underactuated underwater vehicles moving horizontally using some inertial quasi-velocities is considered. The control scheme consists of a backstepping technique along with integral sliding mode control with IQV consideration. These quasi-velocities, resulting from the transformation of the equations of motion in velocity space, are crucial to the control process because the other methods have already been used many times. The feature that distinguishes the developed control scheme from others based on a combination of the aforementioned methods is the fact that the model parameters are used in the dynamic controller and thus the dynamic couplings. This is not just the well-known kinematic transformation because, as a result of the decomposition of the inertia matrix, the obtained IQV contain both kinematic and dynamic parameters. The stability analysis conducted in the sense of Lyapunov showed the feasibility and limitations in the applicability of the presented algorithm.

Simulations were carried out taking into account the actual thrusters and velocity limits of the selected vehicles. The obtained numerical results, obtained for two vehicles with different dynamics, show that the proposed tracking control algorithm can effectively guide underwater vehicles to the desired trajectory and is robust to perturbations of model parameters and selected external disturbances functions. In addition, it is found that the proposed algorithm achieves better control performance than the chosen control scheme in tracking the trajectory of an underwater vehicle moving in the horizontal plane, given the assumed operational conditions.

Based on the results obtained (theoretical analysis and simulation results), it can be concluded that the transformation of variables alone does not guarantee effective performance under different conditions, but should be incorporated into a control scheme that guarantees the expected performance. In turn, the transformation of variables should either improve this performance under certain criteria, or give some insight into the dynamics of the vehicle being regulated.

Some comments are also made on the proposed algorithm using IQV in comparison with other control schemes. Further work may extend the results obtained to other vehicles, as the developed approach can also be applied to other types of underactuated systems for which it is possible to describe the dynamics using a symmetric inertia matrix.

**Funding:** The work was supported by Poznan University of Technology Grant No. 0211/SBAD/0122.

**Institutional Review Board Statement:** Not applicable.

**Informed Consent Statement:** Not applicable.

**Data Availability Statement:** Not applicable.

**Conflicts of Interest:** The author declares no conflict of interest.

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
