# Peer review of "A Quasi-Velocity-Based Tracking Controller for a Class of Underactuated Marine Vehicles"

_applsci, doi:10.3390/app12178903_

Round 1

Reviewer 1 Report

The study is an alternative method for path tracking. This current path needs to be compared with track's methods. For example, a comparison with the results of methods such as stanley control or linear-quadratic regulator in the same simulator will reveal scientific contribution. I suggest you review the following examples:

-https://doi.org/10.1007/s13369-020-04784-0

https://doi.org/10.1016/j.trip.2021.100314

- https://doi.org/10.1177%2F1729881420974852

Author Response

Thank you very much for your review. I am enclosing my responses to the issues indicated.

Reviewer 2 Report

Journal Name: Applied Sciences
Title: A quasi-velocity based tracking controller for a class of underactuated marine vehicles
Article accepted in present form or with minor corrections.

1. How the current work is different from the previous work of the same author "Application of a Trajectory Tracking Algorithm for Underactuated Underwater Vehicles Using Quasi-Velocities" has to be discussed
2. Literature review has to further increased
3. Conclusion has to be improved
4. The size of all subfigures has to be increased
5. The proper distance has to be maintained beteeen axis label and axis tics
6. Please check the grammatical and syntax error

Author Response

(The authors gave the same response as above.)

Reviewer 3 Report

1 For the decomposition method of symmetric inertia matrix M, why use the method in [31], and what advantages does this method have over other methods? Please supplement.

2 This paper only carries out simple simulation for the proposed method without experimental verification. It is difficult to prove the effectiveness of this method in practice and lacks practical significance. It is suggested to increase experiments.

3 The author needs to describe the innovation of this study in detail and compare it with the existing research results. 

4 English abbreviations are only used for the first time and do not need to be repeated. For example, neural network (NN) is repeated many times. Please check the full manuscript.

5 The positions of the serial numbers of equations (5) and (27) should be correctly marked. Equation (32) is mis-written and the transpose symbol is missing. The variables or constraints of the equation need to be explained, such as ρk. Equations (6), (7), and (8) are repeated with the description in the main text. Equations (55) (56) assume that the desired track position profile is two Pd, which should be written as Pd1 and Pd2.

6 The case of equation letters in the text is not uniform, such as ρK and ρk. It needs to be checked and revised throughout.

7 The format of the manuscript is wrong, and the number of lines in the text is missing, as shown in Chapter 4.1. 

8 The format of references is incorrect, such as 3, 5, 36, which should be checked and unified. 

9 The quality of English needs to be improved, especially in the introduction.

10. In the abstract The proposed scheme was verified on two 3 DOF models of underwater vehicles taking into consideration taking into account the limitations of the thrusters, taking into is used repeatedly, and it is suggested to delete one.

Author Response

(The authors gave the same response as above.)

Reviewer 4 Report

In their contribution, the authors focus on the design of a quasi-velocity based tracking controller for a class of underactuated arine vehicles. In general, an underwater vehicle is modeled with 6 degrees of freedom (DOF). In this work, however, a vehicle model with 3 DOF in horizontal motion is considered. The proposed scheme was verified on two 3 DOF models of underwater vehicles taking into consideration taking into account the limitations of the thrusters. I think this post deserves to be published and could be of benefit to colleagues dealing with the same or similar issues.

Author Response

(The authors gave the same response as above.)

Round 2

Reviewer 1 Report

Paper can be accepted in present form